# Association Between Unmet Healthcare Needs and Depression in Older Adults: Evidence from the Korean National Health and Nutrition Examination Survey

**DOI:** 10.3390/healthcare13202635

**Published:** 2025-10-20

**Authors:** Ji-Woo Seok, Kahye Kim, Jaeuk U. Kim, Mi Hong Yim

**Affiliations:** 1Digital Health Research Division, Korea Institute of Oriental Medicine, Yuseong-daero 1672, Yuseong-gu, Daejeon 34054, Republic of Korea; suk6124@kiom.re.kr (J.-W.S.); kkh2@kiom.re.kr (K.K.); jaeukkim@kiom.re.kr (J.U.K.); 2KM Convergence Science, University of Science and Technology, Gajeong-ro 217, Yuseong-gu, Daejeon 34113, Republic of Korea

**Keywords:** depression, unmet healthcare needs, KNHANES, cross-sectional study, older, adults

## Abstract

**Background:** Studies on the relationship between unmet healthcare needs and depression in older adults are limited in Asia. This study aimed to investigate the relationship between unmet healthcare needs and the risk of depression in older adults aged ≥65 years in Korea. **Methods:** This cross-sectional study used data from the Korean National Health and Nutrition Examination Survey in 2014, 2016, and 2018. Three models were constructed using a weighted multivariate logistic regression analysis to account for the complex survey design. Model 1 is adjusted for sex and age. Model 2 was further adjusted for household income, education level, marital status, and private health insurance. Model 3 was further adjusted for alcohol use, cigarette use, weekly walking activity, hypertension, dyslipidemia, and diabetes mellitus. Subgroup analyses were performed to assess the stability and robustness of the association between unmet healthcare needs and depression. **Results:** In total, this study included 4062 participants aged ≥65 years, with 3749 and 313 individuals in the non-depression and depression groups, respectively. In the unadjusted models, individuals with unmet healthcare needs had a greater likelihood of depression than those without unmet needs. This association remained significant across all three sequentially adjusted models. Subgroup analyses supported these findings. **Conclusions:** Unmet healthcare needs are significantly associated with depressive symptoms in older adults. Therefore, a multidimensional approach is required when addressing mental health issues in this population, and addressing unmet healthcare needs may be crucial for promoting mental health.

## 1. Introduction

The aging global population represents more than a demographic shift; it highlights a range of health challenges for older adults, making them a significant social priority [1]. Older adults are particularly vulnerable to depression due to multiple factors, including biological aging, chronic diseases, functional decline, social isolation, bereavement, and economic hardship [2,3,4]. Yet, only about 20% of older adults with depression receive treatment, and untreated cases can lead to severe outcomes such as suicide, worsening comorbidities, and recurrent episodes [5,6,7,8].

Recent meta-analyses estimate the global prevalence of depression among older adults at 13.3–28.4% [9,10]. However, prevalence differs across countries, reflecting variations in assessment tools, populations, and sociocultural contexts. In the United States, the lifetime and 12-month prevalence of major depressive disorder (MDD) have been reported as 8.2% and 2.7% [11]. In China, the estimates range from 1.5% to 7.9% [12]. In Korea, the prevalence is roughly 2–4 percentage points higher than in other Asian countries, a trend likely related to cultural norms, family structure changes, economic instability, and unequal healthcare access [5].

Beyond its clinical features, depression reflects multidimensional vulnerability in older adults [13]. It is often accompanied by malnutrition, chronic pain, sleep disturbances [14], reduced quality of life [15], social isolation [16], and poverty with limited access to resources [17,18]. The cumulative impact of these disadvantages substantially elevates depression risk [13].

This vulnerability is also captured by unmet healthcare needs [13,19]. Unmet healthcare needs occur when individuals cannot obtain necessary medical services [20]. They are shaped not only by healthcare system accessibility but also by economic conditions, transportation, social support, and health literacy [21]. Older adults face heightened risk due to physical decline, chronic disease burden, financial strain, and social isolation [22,23,24,25]. Prior studies show that unmet needs are associated with worse health outcomes, poor chronic disease management, reduced quality of life, and premature mortality [19,26].

Most research has conceptualized depression as a predictor of unmet needs [27]. By contrast, fewer studies have examined whether unmet healthcare needs themselves contribute to depression. Evidence on this relationship is limited and often based on cross-sectional data or restricted populations [26,28]. Nevertheless, significant associations have been reported. For example, in France, unmet healthcare needs were more prevalent in those with depression (37.2% vs. 19.3%; aOR = 1.66, 95% CI: 1.27–2.18) [29]. In China, the prevalence of unmet needs was 13.0%, and depressive symptoms were significantly associated with higher odds of unmet needs [30]. In Korea, 17.4% of older adults reported unmet needs, and depressive symptoms increased the risk 1.45-fold [31].

Unmet healthcare needs may thus extend beyond delayed treatment, contributing to psychosocial problems such as declining health, reduced self-esteem, and helplessness [28,32]. Consistently, older adults with unmet needs show poorer psychological health and higher depression risk [32,33,34,35]. Evidence from diverse populations, including low-income groups, German and Malaysian cohorts, and European multicountry samples, supports a direct link between unmet needs and depressive symptoms [32,33,35,36].

Taken together, these findings suggest that unmet healthcare needs may act as an independent risk factor for depression, especially when combined with vulnerabilities such as socioeconomic disadvantage, poor health behaviors, and social isolation [23,24,25,27,32,33]. Yet, large-scale population-based studies that rigorously test these interactions remain scarce [32,33]. In addition, the association between unmet healthcare needs and depression may differ by sociodemographic characteristics such as sex, income, and marital status, which are closely tied to cultural and institutional contexts. Given Korea’s rapid demographic transition [5], weakening family support [16], relatively high out-of-pocket medical costs despite universal health insurance [21], and strong cultural stigma toward mental illness, investigating this issue in Korea provides insights that cannot be directly inferred from studies conducted in Europe or North America.

This study addressed the particularly high prevalence of depression among older adults in Korea [5]. Because the socioeconomic and cultural contexts of older Koreans differ from those of other Asian populations, it is important to assess whether unmet healthcare needs remain associated with depression beyond these factors. Using nationally representative data from the 2014, 2016, and 2018 Korean National Health and Nutrition Examination Survey (KNHANES) [37], this study investigated whether unmet healthcare needs function as an independent risk factor for late-life depression. Findings are expected to inform the identification of high-risk groups and guide interventions to reduce mental health disparities in Korea’s aging population.

## 2. Methods

### 2.1. Data Sources and Study Participants

This cross-sectional study used data from the KNHANES, conducted by the Korea Disease Control and Prevention Agency in 2014, 2016, and 2018. The KNHANES is a large-scale national survey focused on health and nutrition, and its sample was selected using a two-stage stratified cluster sampling method to ensure the representativeness and reliability of the entire Korean population [37,38]. The KNHANES comprises a health interview survey that investigates disease prevalence, healthcare utilization, sociodemographic characteristics, and health behaviors; a health examination that includes anthropometric measurements, blood pressure and pulse measurements, and blood and urine tests; and a nutrition survey that examines dietary behavior, food intake, and other related factors [37,38]. The health interview survey and health examination were conducted at mobile examination centers, whereas the nutrition survey was conducted through direct visits to participant households. Depending on the survey items, data were collected using either interviewer- or self-administered methods. Raw data from KNHANES are provided after anonymization to prevent the identification of individuals, in compliance with the Personal Information Protection Act and Statistics Act, and can be accessed from the official KNHANES website after signing and submitting the Terms of Use Agreement for Statistical Data Users and the Confidentiality Agreement (https://knhanes.kdca.go.kr/knhanes/main.do accessed on 1 March 2025). This study was approved for exemption from review by the Institutional Review Board (IRB) of the Korea Institute of Oriental Medicine as it involved the secondary use of previously de-identified data (IRB No., I-2408/008-001).

We used KNHANES data from 2014, 2016, and 2018, during which the PHQ, a tool for assessing depressive symptoms, was implemented. In total, 23,692 individuals participated in KNHANES. Among them, 1072 participants who did not participate in the health interview survey, 17,906 aged <65 years, 359 who did not require healthcare within the past year, 161 with missing PHQ responses, and 132 with missing values for variables such as education level, household income, marital status, private health insurance coverage, alcohol use, cigarette use, and weekly walking activity were excluded. Finally, 4062 participants were included in this study: 1778 men and 2284 women (Figure 1).

### 2.2. Assessment of Depressive Symptoms

Depressive symptoms were assessed using PHQ-9, a depression screening tool comprising nine items that measure the frequency of depressive symptoms. Each item is scored from 0 (not at all) to 3 (nearly every day), resulting in a total score ranging from 0 to 27 across the nine items, with higher scores indicating a greater risk of depression [39]. Information on the PHQ-9 has been collected via self-administered methods every 2 years since 2014, following item-by-item instructions from the standardized protocol in the KNHANES. Participants were asked how often they experienced the symptoms of each PHQ-9 item in the past 2 weeks. Individuals with a total PHQ-9 score of ≥10 and <10 were classified into the depression non-depression groups, respectively [39]. This criterion for identifying depression has been widely used in clinical and epidemiological studies, and its performance has been validated [39,40].

### 2.3. Assessment of Unmet Healthcare Needs

Unmet healthcare needs were determined based on the questionnaire item, “During the past year, have you ever needed medical care, including examinations or treatments, at medical institutions (excluding dental services), but been unable to receive it?” Individuals who answered yes were classified as having unmet healthcare needs, whereas those who answered no were classified as not having unmet healthcare needs. Individuals who reported not needing medical care during the past year were excluded. Responses to the questionnaire items were collected using an interviewer-administered method by a well-trained interviewer following a standardized protocol.

### 2.4. Assessment of Covariates

Sociodemographic characteristics, health behaviors, and chronic diseases were considered as covariates to evaluate the association between unmet healthcare needs and depression. Sociodemographic characteristics and chronic diseases were gathered through interviewer-administered methods, whereas health behaviors were obtained through self-administered methods, following a standardized protocol. Sociodemographic characteristics included sex (men and women), age (65–74 and ≥75 years), household income (1st quartile [lowest], 2nd quartile, and 3rd–4th quartile [highest]), education level (elementary school or below, middle school, and high school or above), marital status (married/living together and widowed/divorced/separated/never married), and private health insurance status (yes or no). Health behaviors included alcohol use (none, monthly or less, and 2 times a month or more), cigarette use (none/quit smoking and occasionally/every day), and weekly walking activity (none, 1–6 days, and every day). Chronic diseases included hypertension (no or yes), dyslipidemia (no or yes), and diabetes mellitus (no or yes).

### 2.5. Statistical Analysis

In all statistical analyses of the KNHANES data, complex sample design elements, such as clustering, stratification, and weighting, were applied to generalize the results to the target Korean population. R version 4.4.1 (R Foundation for Statistical Computing, Vienna, Austria) was used to preprocess the data, and the complex samples procedure in IBM SPSS Statistics for Windows (version 29.0; IBM Corp., Armonk, NY, USA) was used to perform statistical modeling and produce weighted estimates. A two-sided hypothesis was used for all statistical tests, and a significance level of 0.05 was applied.

To compare general participant characteristics between the non-depression and depression groups in adults aged ≥65 years, chi-square tests with Rao-Scott second-order correction were applied to account for the complex survey design for categorical variables. The results were summarized as unweighted frequencies and weighted column percentages [41]. Similarly, to compare participant characteristics according to met and unmet healthcare needs in adults aged ≥65 years, the statistical methods and results were consistent with those used to compare general characteristics between the non-depression and depression groups. To identify the association between unmet healthcare needs and depression in adults aged ≥65 years, weighted logistic regression analyses were performed. For the unadjusted analysis, a weighted univariate logistic regression analysis was conducted. For the adjusted analysis, three models were constructed using weighted multivariate logistic regression. Model 1 is adjusted for sex and age. Model 2 was further adjusted for household income, education level, marital status, and private health insurance in addition to the covariates in Model 1. Model 3 was further adjusted for alcohol use, cigarette use, weekly walking activity, hypertension, dyslipidemia, and diabetes mellitus in addition to the covariates in Model 2. This hierarchical model was applied by sequentially adding covariates, which made it possible to observe how the association between unmet healthcare needs and depression changed as demographic, socioeconomic, and health-related factors were considered. The results were reported as odds ratios (OR) and 95% confidence intervals (CI). Subgroup analyses and interaction tests were performed to further assess the stability and robustness of the association between unmet healthcare needs and depression. Subgroups were stratified according to sex, age, household income, education level, marital status, private health insurance, alcohol use, cigarette use, weekly walking activity, hypertension, dyslipidemia, and diabetes mellitus. For each subgroup, weighted multivariate logistic regression models were fitted, adjusting for all covariates except the stratification variable under examination. Further, the interaction effects between unmet healthcare needs and each stratification variable were analyzed. The results are presented as adjusted ORs (aORs) with 95% CI and p value for interaction effects.

## 3. Results

### 3.1. General Participant Characteristics Between the Non-Depression and Depression Groups

In total, 4062 participants aged ≥65 years were included in this study, with 3749 (weighted proportion, 92.63%) and 313 individuals (7.37%) in the non-depression and depression groups, respectively. Significant differences were observed between both groups regarding unmet healthcare needs, sex, household income, educational level, marital status, private health insurance, alcohol use, weekly walking activities, dyslipidemia, and diabetes mellitus. The depression group had higher proportions of unmet healthcare needs (36.18% vs. 8.85%), women (78.98% vs. 55.37%), household income of lowest quartile (67.22% vs. 45.51%), education level of elementary school or below (77.02% vs. 56.15%), marital status of widowed/divorced/separated/never married (53.65% vs. 32.93%), no private health insurance (69.74% vs. 56.8%), no alcohol use (63.38% vs. 47.22%), no weekly walking activity (44.42% vs. 24.68%), presence of dyslipidemia (34.52% vs. 25.25%), and presence of diabetes mellitus (29.11% vs. 20.48%) than the non-depression group (Table 1).

### 3.2. Participant Characteristics by Met and Unmet Healthcare Needs

Among the 4062 participants aged ≥65 years who reported needing medical care during the past year, 438 individuals (10.86%) had unmet healthcare needs, whereas 3624 individuals (89.14%) had their healthcare needs met. Significant differences were observed between the met and unmet healthcare needs groups regarding depression, sex, household income, education level, marital status, private health insurance, alcohol use, and weekly walking activity. The unmet healthcare needs group had higher proportions than the met healthcare needs group in the following characteristics: presence of depression (24.55% vs. 5.28%), women (75.28% vs. 54.89%), household income of the lowest quartile (60.20% vs. 45.51%), education level of elementary school or below (75.14% vs. 55.56%), marital status of widowed/divorced/separated/never married (49.21% vs. 32.66%), no private health insurance (63.90% vs. 57.00%), no alcohol use (54.79% vs. 47.63%), and no weekly walking activity (37.08% vs. 24.81%) (Table 2).

### 3.3. Association Between Unmet Healthcare Needs and Depression

In the unadjusted model, a significant association was observed between unmet healthcare needs and depression in adults aged ≥65 years. Individuals with unmet healthcare needs were more likely to have depression than those without unmet needs (unadjusted OR [95% CI], 5.84 [4.35–7.85]). Even across the three models, with sequential adjustments for sex and age, additional sociodemographic variables, and health-related factors such as health behaviors, chronic diseases, and unmet healthcare needs remained significantly associated with depression. In Model 1, adjusted for sex and age, individuals with unmet healthcare needs had a greater likelihood of depression than those with met needs (aOR [95% CI], 5.07 [3.70–6.95]). In Model 2, which included additional adjustments for sociodemographic variables (household income, education level, marital status, and private health insurance) beyond Model 1, individuals with unmet healthcare needs had a higher likelihood of depression than those without unmet needs (4.44 [3.24–6.10]). In Model 3, which included additional adjustments for health-related factors (alcohol use, cigarette use, weekly walking activity, hypertension, dyslipidemia, and diabetes mellitus) beyond Model 2, individuals with unmet healthcare needs were more likely to have depression than those without unmet needs (4.3 [3.12, 5.93]) (Table 3).

### 3.4. Association Between Unmet Healthcare Needs and Depression Stratified by Subgroups

Subgroup analyses were conducted to evaluate the stability and robustness of the associations between unmet healthcare needs and depression. Subgroups were stratified by each covariate, adjusting for all covariates except for the stratification variable under examination. In all subgroups except for those in the highest household income quartile, unmet healthcare needs were significantly associated with a higher probability of depression: individuals with unmet healthcare needs were more likely to experience depression than those without unmet needs (*p* values for aOR in each subgroup < 0.05). These findings were consistent with the results of the main analysis without stratification. In addition, the interaction effects between unmet healthcare needs and each stratification variable were analyzed; however, no significant interactions were observed (*p* > 0.05), indicating that the association between unmet healthcare needs and depression remained consistent across all subgroups (Figure 2).

## 4. Discussion

This study identified a relationship between experiences of unmet healthcare needs and depressive symptoms in an older population. The high-risk population for depression was characterized by a PHQ-9 score of ≥10. The results showed that individuals who experienced unmet healthcare needs had a substantially higher risk of depression than those who did not experience unmet needs. This relationship remained statistically significant even after adjusting for major confounding variables, such as sex, age, marital status, education level, household income, private health insurance, health behavior, and chronic diseases. Subgroup analyses indicated a significant correlation between unmet healthcare needs and depressive symptoms across most sociodemographic characteristics. Notably, significant results were observed in all income subgroups except for those in the top 25% of household income. However, tests for interaction revealed no significant differences in the strength of this correlation across subgroups, indicating that unmet healthcare needs are not confined to a particular demographic but instead represent a broadly applicable risk factor that adversely affects mental health across the older population.

Our findings suggest that unmet healthcare needs may serve as a substantial and enduring risk factor for depressive symptoms in older adults, extending beyond delays or inconveniences in accessing medical services.

A cross-sectional study of Korean older adults reported that individuals with depressive symptoms were 1.45 times more likely to have unmet healthcare needs than those without depressive symptoms [31]. A study using data from the China Health and Retirement Longitudinal Survey revealed that insufficient medical and emotional support causes more severe depressive symptoms among older adults [22]. Stein et al. (2019) indicated that in nations with inadequate mental health management systems, disparities in access to medical care are more likely to result in undetected or untreated late-life depression, significantly impairing the quality of life among older adults [35]. Consistent with previous studies, this study identified unmet healthcare experiences as a significant predictor of a high-risk group for depression (PHQ-9 ≥ 10) among older adults (OR = 4.30). Although older adults regularly seek care in medical settings, they often encounter various barriers that hinder their access to adequate mental health services, arising from structural barriers, including challenges in communicating with medical providers, stigma associated with mental illness, and limited access to services [26,33,35,42]. The recurrence of unfulfilled healthcare experiences may diminish treatment expectations and exacerbate feelings of emotional isolation and helplessness. In line with previous studies, such unmet needs may also contribute to depression through broader psychosocial pathways, including deterioration of overall health, reduced self-esteem, and heightened feelings of helplessness [27,31]. Moreover, the relationship between unmet healthcare needs and depressive symptoms may be reciprocal rather than unidirectional, as depression itself can further limit healthcare-seeking behaviors.

The descriptive results suggest that unmet healthcare needs are not merely isolated incidents but rather multifaceted issues closely linked to the socioeconomic and health behavioral vulnerabilities prevalent among the high-risk population for depression. Table 1 shows that the high-risk depression group exhibited significantly higher rates among women, individuals with low income, those who were elementary school graduates or below education, unmarried individuals (including divorced, widowed, separated, or never married), non-subscribers to private health insurance, and those lacking physical activity such as walking, suggesting that individuals in the high-risk depression group face not only emotional difficulties, but also structural vulnerabilities across multiple domains, including socioeconomic status, health behaviors, and social isolation. This pattern is also evident in Table 2. Individuals with unmet healthcare needs not only had a higher prevalence of depressive symptoms but also demonstrated overall disadvantages in health and social conditions, including low income, low education, being unmarried, no insurance coverage, unhealthy behavior, and lack of physical activity, suggesting that unmet healthcare needs are not merely the result of service delivery failure but may reflect structural problems wherein individual multidimensional vulnerabilities interact in a complex manner.

Previous studies conducted in various countries have reported this pattern. A study involving an older population in Turkey revealed that factors such as self-perceived low-income status, unmarried status, and lack of health insurance coverage were significantly correlated with unmet healthcare needs [43]. A study with a large sample of Chinese adults aged ≥60 years found that unmet healthcare needs were associated with a two-fold increase in the likelihood of experiencing depressive symptoms, especially in those with poorer health conditions, low income, and limited education [44]. A study using a French cross-sectional survey reported that 23.0% of older adults in France experienced at least one type of unmet healthcare need. Unmet healthcare needs were significantly associated with depressive symptoms, with the highest risk observed among those aged ≥90 years, individuals with perceived economic hardship, and limitations in instrumental activities of daily living [29]. Furthermore, a study analyzing the Survey of Health, Aging, and Retirement in Europe data across 12 European nations revealed that unmet healthcare needs frequently co-occur with factors such as living alone, low income, limited education, and daily activity limitations [32], consistent with the vulnerability traits identified in the high-risk group for depression in our study. These findings, repeatedly observed across diverse cultural and social contexts, suggest that unmet healthcare needs may not simply reflect the temporary limitations of the healthcare system but rather act as a complex social risk factor. Address mental health issues among the older population requires diagnosis and intervention within the multidimensional context of social isolation, economic limitations, and lack of health behaviors rather than solely relying on a technical approach that improves accessibility to medical care.

The results of the subgroup analysis showed that unmet healthcare needs were significantly associated with depressive symptoms across diverse demographic groups (Figure 2). In particular, the aORs were significant in all subgroups except the top 25% income group, suggesting that restricted access to health care may be a crucial risk factor for depressive symptoms in the older population. To more precisely evaluate the difference in the strength of the correlation across subgroups, an interaction-effect analysis was performed. No significant interaction effects were observed for any variable, including sex, age, education level, income level, private insurance, and health behavior, suggesting that the association between unmet healthcare needs and depression shows a consistent pattern across groups, with no clear evidence that the relationship is substantially stronger or weaker in any particular subgroup. This indicates that unmet medical needs are consistently associated with depression in older adults regardless of individual characteristics. Although the severity of these effects may vary according to socioeconomic status, they remain fundamental factors in mental health vulnerability, indicating that unmet healthcare needs may have a wide-ranging and consistent negative impact on the emotional well-being of older adults beyond merely a lack of access to medical services [32]. The magnitude of this association may differ depending on economic resource availability; however, fundamentally, limited access to medical care is likely to exacerbate psychosocial vulnerability in older adults.

Notably, no significant association was found between unmet healthcare needs and depression in the top 25% income group. This finding aligns with that of prior studies indicating that economic resources help mitigate emotional distress in individuals facing stressful situations [45,46,47]. Studies have reported that economic resources may contribute to preventing depressive symptoms by improving access to health-promoting resources, reducing chronic stressors associated with financial strain, and providing individuals with greater control over their environment and life circumstances [48,49]. In addition, groups with fewer economic resources are reportedly more vulnerable to stress and have lower resilience, whereas groups with more economic resources tend to have a greater capacity to absorb economic and psychological shocks [46,50,51]. The findings suggest that in formulating policies aimed at improving healthcare accessibility in the future, considering not only increasing overall utilization rates but also varying protective effects based on the level of resource availability is essential.

This study has several limitations. First, the analysis relies on cross-sectional data, which precludes the establishment of causality. It remains plausible that depression could contribute to unmet healthcare needs, rather than solely resulting from them. Second, depressive symptoms were assessed using a self-report questionnaire (PHQ-9); thus, they cannot fully substitute for a clinical diagnosis. Third, unmet healthcare needs were measured based on respondents’ subjective perceptions and may therefore be influenced by recall or perceptual bias. Fourth, despite adjusting for multiple covariates, the possibility of residual confounding cannot be entirely ruled out. Fifth, this study utilized data from the 2014, 2016, and 2018 waves of the KNHANES because, at the time of analysis, the 2022 and 2024 datasets were not yet publicly available, and PHQ-9 data are collected only biennially. The 2020 wave was excluded, as it coincided with the COVID-19 pandemic, during which government policies restricted hospital visits for non-critical cases. These exceptional circumstances may have substantially altered both unmet healthcare needs and depressive symptoms, and thus were deemed inappropriate for inclusion in this analysis. Although the data were collected before the COVID-19 pandemic, they provide important baseline information on the pre-pandemic status of unmet healthcare needs and depression among older adults in Korea. Once the most recent KNHANES datasets (2022 and 2024 waves) become publicly available, our research team plans to conduct a time-trend analysis using the same national big data source to compare patterns observed before, during, and after the pandemic. This future analysis will address a distinct research question and will therefore be presented as an independent study. Finally, this study focused on Korean older adults (≥65 years) within the context of a unique national health insurance system and demographic structure. Therefore, caution is warranted when generalizing these findings to younger age groups or to populations in different sociocultural and healthcare settings.

## 5. Conclusions

In conclusion, this study found a robust association between unmet healthcare needs and depressive symptoms among older adults across most sociodemographic subgroups. The persistence of this relationship despite Korea’s high healthcare accessibility suggests that unmet needs are not solely economic but also reflect non-economic vulnerabilities such as social isolation and activity limitations. Although the association was not significant in the highest-income group, likely reflecting the buffering role of economic resources, unmet healthcare needs overall emerged as an independent, multidimensional risk factor for late-life depression. These findings highlight the need for multidimensional approaches that go beyond improving service availability, including early identification of high-risk groups, preventive strategies, and interventions tailored to socioeconomic and social vulnerabilities to reduce mental health disparities in aging populations.

## Figures and Tables

**Figure 1 healthcare-13-02635-f001:**
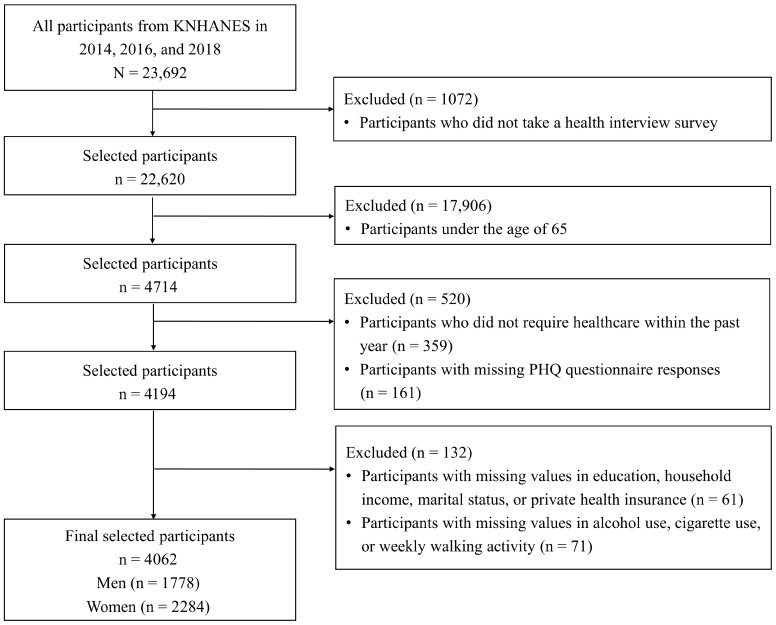
Flowchart of study participant selection.

**Figure 2 healthcare-13-02635-f002:**
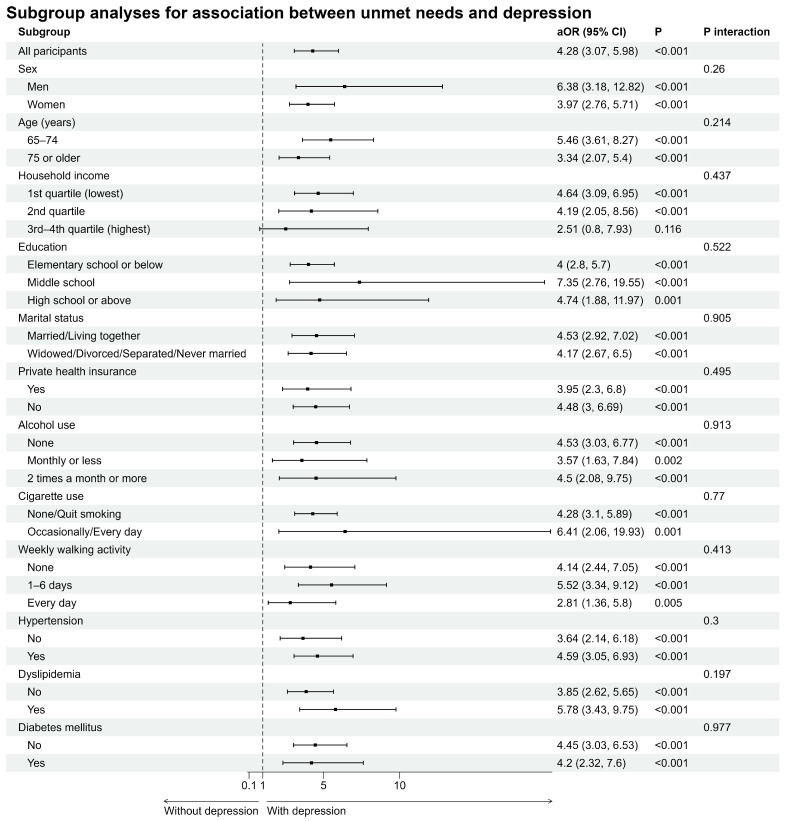
Subgroup analyses for association between unmet needs and depression among adults aged 65 years and older. Subgroup were stratified by sex, age, household income, education level, marital status, private health insurance, alcohol use, cigarette use, weekly walking activity, hypertension, dyslipidemia, and diabetes mellitus. Each analysis was adjusted for all covariates except the stratification variable itself. Weighted multivariable logistic regression analyses were performed for each subgroup. The results were presented as adjusted odds ratios with 95% confidence interval and *p* value for interaction effects between unmet needs for healthcare and each stratification variable. All analyses accounted for the complex survey design. Color is used only to improve readability.

**Table 1 healthcare-13-02635-t001:** General participant characteristics between non-depression and depression groups.

Variables	Overall	Non-Depression	Depression	*p* Value
Number of participants	4062	3749	313	
Unmet needs				<0.001
No	3624 (89.14)	3415 (91.15)	209 (63.82)	
Yes	438 (10.86)	334 (8.85)	104 (36.18)	
Sex				<0.001
Men	1778 (42.89)	1704 (44.63)	74 (21.02)	
Women	2284 (57.11)	2045 (55.37)	239 (78.98)	
Age (years)				0.256
65–74	2541 (60.74)	2355 (61.02)	186 (57.21)	
≥75	1521 (39.26)	1394 (38.98)	127 (42.79)	
Household income				<0.001
1st quartile (lowest)	1916 (47.11)	1705 (45.51)	211 (67.22)	
2nd quartile	1089 (26.14)	1017 (26.37)	72 (23.18)	
3rd- 4th quartile (highest)	1057 (26.76)	1027 (28.12)	30 (9.6)	
Education				<0.001
Elementary school or below	2360 (57.69)	2120 (56.15)	240 (77.02)	
Middle school	623 (15.21)	587 (15.5)	36 (11.58)	
High school or above	1079 (27.1)	1042 (28.35)	37 (11.4)	
Marital status				<0.001
Married/Living together	2755 (65.54)	2594 (67.07)	161 (46.35)	
Widowed/Divorced/Separated/Never married	1307 (34.46)	1155 (32.93)	152 (53.65)	
Private health insurance				<0.001
Yes	1677 (42.25)	1581 (43.2)	96 (30.26)	
No	2385 (57.75)	2168 (56.8)	217 (69.74)	
Alcohol use				<0.001
None	1970 (48.41)	1776 (47.22)	194 (63.38)	
Monthly or less	931 (22.92)	877 (23.42)	54 (16.63)	
2 times a month or more	1161 (28.67)	1096 (29.36)	65 (19.99)	
Cigarette use				0.842
None/Quit smoking	3684 (90.93)	3400 (90.96)	284 (90.56)	
Occasionally/Every day	378 (9.07)	349 (9.04)	29 (9.44)	
Weekly walking activity				<0.001
None	1089 (26.14)	956 (24.68)	133 (44.42)	
1–6 days	1720 (41.98)	1613 (42.66)	107 (33.49)	
Every day	1253 (31.88)	1180 (32.66)	73 (22.09)	
Hypertension				0.064
No	1879 (46.03)	1750 (46.47)	129 (40.44)	
Yes	2183 (53.97)	1999 (53.53)	184 (59.56)	
Dyslipidemia				0.003
No	2994 (74.06)	2786 (74.75)	208 (65.48)	
Yes	1068 (25.94)	963 (25.25)	105 (34.52)	
Diabetes mellitus				0.006
No	3203 (78.88)	2977 (79.52)	226 (70.89)	
Yes	859 (21.12)	772 (20.48)	87 (29.11)	

Abbreviations: BMI, Body mass index. *p*-values were derived using chi-square tests with the Rao-Scott second-order correction to account for the complex survey design of the categorical variables. The results are summarized as unweighted frequencies and weighted column percentages.

**Table 2 healthcare-13-02635-t002:** Participant characteristics by met and unmet needs.

Variables	Overall	Met Needs	Unmet Needs	*p* Value
Number of participants	4062	3624	438	
Depression				<0.001
No	3749 (92.63)	3415 (94.72)	334 (75.45)	
Yes	313 (7.37)	209 (5.28)	104 (24.55)	
Sex				<0.001
Men	1778 (42.89)	1667 (45.11)	111 (24.72)	
Women	2284 (57.11)	1957 (54.89)	327 (75.28)	
Age (years)				0.101
65–74	2541 (60.74)	2286 (61.34)	255 (55.79)	
≥75	1521 (39.26)	1338 (38.66)	183 (44.21)	
Household income				<0.001
1st quartile (lowest)	1916 (47.11)	1650 (45.51)	266 (60.2)	
2nd quartile	1089 (26.14)	988 (26.41)	101 (23.88)	
3rd–4th quartile (highest)	1057 (26.76)	986 (28.08)	71 (15.93)	
Education				<0.001
Elementary school or below	2360 (57.69)	2030 (55.56)	330 (75.14)	
Middle school	623 (15.21)	572 (15.65)	51 (11.59)	
High school or above	1079 (27.1)	1022 (28.79)	57 (13.26)	
Marital status				<0.001
Married/Living together	2755 (65.54)	2517 (67.34)	238 (50.79)	
Widowed/Divorced/Separated/Never married	1307 (34.46)	1107 (32.66)	200 (49.21)	
Private health insurance				0.018
Yes	1677 (42.25)	1521 (43)	156 (36.1)	
No	2385 (57.75)	2103 (57)	282 (63.9)	
Alcohol use				0.002
None	1970 (48.41)	1724 (47.63)	246 (54.79)	
Monthly or less	931 (22.92)	828 (22.7)	103 (24.75)	
2 times a month or more	1161 (28.67)	1072 (29.67)	89 (20.46)	
Cigarette use				0.553
None/Quit smoking	3684 (90.93)	3285 (90.83)	399 (91.78)	
Occasionally/Every day	378 (9.07)	339 (9.17)	39 (8.22)	
Weekly walking activity				<0.001
None	1089 (26.14)	923 (24.81)	166 (37.08)	
1–6 days	1720 (41.98)	1545 (42.28)	175 (39.57)	
Every day	1253 (31.88)	1156 (32.92)	97 (23.35)	
Hypertension				0.417
No	1879 (46.03)	1692 (46.3)	187 (43.82)	
Yes	2183 (53.97)	1932 (53.7)	251 (56.18)	
Dyslipidemia				0.16
No	2994 (74.06)	2689 (74.46)	305 (70.84)	
Yes	1068 (25.94)	935 (25.54)	133 (29.16)	
Diabetes mellitus				0.945
No	3203 (78.88)	2864 (78.87)	339 (79.02)	
Yes	859 (21.12)	760 (21.13)	99 (20.98)	

Abbreviations: BMI, Body mass index. Details for calculating *p*-values and expressing results are consistent with those outlined in Table 1.

**Table 3 healthcare-13-02635-t003:** Association between unmet needs and depression in adults aged ≥65 years.

	Unadjusted Model		Model 1		Model 2		Model 3	
	uOR (95% CI)	*p* Value	aOR (95% CI)	*p* Value	aOR (95% CI)	*p* Value	aOR (95% CI)	*p* Value
Unmet needs								
No	1.00 (ref.)	NA	1.00 (ref.)	NA	1.00 (ref.)	NA	1.00 (ref.)	NA
Yes	5.84 (4.35, 7.85)	<0.001	5.07 (3.7, 6.95)	<0.001	4.44 (3.24, 6.10)	<0.001	4.30 (3.12, 5.93)	<0.001

Abbreviations: aOR, adjusted odds ratio; CI, confidence interval; uOR, unadjusted odds ratio; ref., reference category; NA, not applicable. Model 1: Adjusted for sex and age. Model 2: Adjusted for covariates in Model 1 plus household income, education level, marital status, and private health insurance. Model 3: Adjusted for covariates in Model 2 plus alcohol use, cigarette use, weekly walking activity, hypertension, dyslipidemia, and diabetes mellitus. *p*-values were obtained using weighted univariate logistic regression analysis for the unadjusted analysis and weighted multivariate logistic regression analysis for the adjusted analysis to account for the complex survey design. The results are reported as odds ratios and 95% confidence intervals.

## Data Availability

Data available in a publicly accessible repository. The original data presented in this study are openly available in Korea National Health and Nutrition Examination Survey (KNHANES) at https://knhanes.kdca.go.kr/knhanes/main.do (accessed on 1 March 2025).

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
