# Peer review of "Association Between Unmet Healthcare Needs and Depression in Older Adults: Evidence from the Korean National Health and Nutrition Examination Survey"

_healthcare, 2025, doi:10.3390/healthcare13202635_

Round 1

Reviewer 1 Report

Comments and Suggestions for Authors

Thank you for inviting me to review this paper. This study explored the association between unmet healthcare needs and depression in older adults utilizing the KNHANES data. The authors employed logistic regressions to find that unmet healthcare needs are associated with depression. The methodology is sound and results are clearly presented. Overall, the paper is generally well-written, and I only have some minor comments for authors. Thank you.

Introduction

The introduction is somewhat lengthy, and some paragraphs may be condensed.

  1. Line 89–112: the methodological considerations, such as which sociodemographic characteristics were adjusted for, or details on regression models– stepwise approach or subgroup analyses-, are not necessary for the introduction. These can be omitted or moved to the methods section.
  2. Are there any studies examining the relations between unmet healthcare needs and depression among older adults?

Results

Table 3: please consistently use two-decimal points across the estimates (e.g. 6.1 -> 6.10)

Discussion/Conclusion

  1. The possibility of reverse causation, in which those with depression may be more likely to be reluctant to seek help or visit healthcare services, should be acknowledged in the limitation/discussion section.
  2. The sentence “However, in most other older adult groups, unmet healthcare needs negatively affect mental health” should be replaced with “~~ negatively associated with depression”

Author Response

Thank you for your review. Please see the attached file for detailed responses.

Reviewer 2 Report

Comments and Suggestions for Authors

Association Between Unmet Healthcare Needs and Depression in Older Adults: Evidence from the Korean National Health and Nutrition Examination Survey

This study examined the relationship between unmet healthcare needs and depression among older adults by analysing data from the Korean National Health and Nutrition Examination Survey

Introduction - in line with the study purpose.

Methodology -well detailed. Appropriate design. Limited sample that excludes the possibility of generalization of results. Very old data.

The objectives are consistent with the results and conclusions

Unfortunately, other sociodemographic factors that could have influenced the results of the study (e.g. healthcare access, family support, caregiving needs and type, residence) are not taken into account.

Very old data – results  can only be considered a snapshot of 2024, 2016, 2018 of South Korean older people’s condition. 

Author Response

(The authors gave the same response as above.)

Reviewer 3 Report

Comments and Suggestions for Authors

Thank you for the opportunity to review the manuscript. The study is well-developed and presented.

  1. The authors state that “most existing evidence is based on studies conducted in Europe or North America.” Please provide justification whether study issue depends on culture that it should be conducted in Asia, particularly in Korea
  2. Please clarify the rationale for using a hierarchical regression model instead of the enter method. What advantages does the hierarchical approach offer, and what is the theoretical basis for classifying variables into three levels?
  3. In the Discussion section, please refrain from including statistical indices (e.g., aOR = 4.30, 95% CI = 2.30–8.03, p < .001).
  4. Please elaborate on how unmet healthcare needs may be associated with depressive symptoms. Any speculation or theoretical insights would be appreciated.
  5. Lastly, please provide the practical implications of the study findings.

Author Response

(The authors gave the same response as above.)

Round 2

Reviewer 2 Report

Comments and Suggestions for Authors

Introduction was improved. Major issues have been addressed . 

I still consider the manuscript is showing very old data, not showing influence of pandemic and post-pandemic period.

Author Response

Introduction was improved. Major issues have been addressed.

I still consider the manuscript is showing very old data, not showing influence of pandemic and post-pandemic period.

Thank you for your insightful comment. As noted, this study utilized data collected before the COVID-19 pandemic and therefore does not reflect changes that occurred during or after the pandemic. However, because the pandemic has brought substantial structural changes across social, healthcare, and psychological domains, combining pre-, during-, and post-pandemic data within a single analytical framework would not be appropriate. Each period represents a distinct social and policy context, and integrating them into one analysis could obscure meaningful temporal differences.

We are currently conducting a separate time-trend analysis using the same national big data source to compare patterns observed before, during, and after the pandemic. Nevertheless, that analysis addresses a different research question and thematic focus; thus, it will be reported as an independent manuscript rather than an extension of the present study. This temporal limitation and the lack of post-pandemic data will be explicitly acknowledged in the Limitations section of the manuscript, so that readers can interpret the findings with an appropriate understanding of the study’s temporal scope.

Limitation in Discussion

“This study has several limitations. First, the analysis relies on cross-sectional data, which precludes the establishment of causality. It remains plausible that depression could contribute to unmet healthcare needs, rather than solely resulting from them. Second, depressive symptoms were assessed using a self-report questionnaire (PHQ-9); thus, they cannot fully substitute for a clinical diagnosis. Third, unmet healthcare needs were measured based on respondents’ subjective perceptions and may therefore be influenced by recall or perceptual bias. Fourth, despite adjusting for multiple covariates, the possibility of residual confounding cannot be entirely ruled out.

Fifth, this study utilized data from the 2014, 2016, and 2018 waves of the KNHANES because, at the time of analysis, the 2022 and 2024 datasets were not yet publicly available, and PHQ-9 data are collected only biennially. The 2020 wave was excluded, as it coincided with the COVID-19 pandemic, during which government policies restricted hospital visits for non-critical cases. These exceptional circumstances may have substantially altered both unmet healthcare needs and depressive symptoms, and thus were deemed inappropriate for inclusion in this analysis. Although the data were collected before the COVID-19 pandemic, they provide important baseline information on the pre-pandemic status of unmet healthcare needs and depression among older adults in Korea. Once the most recent KNHANES datasets (2022 and 2024 waves) become publicly available, our research team plans to conduct a time-trend analysis using the same national big data source to compare patterns observed before, during, and after the pandemic. This future analysis will address a distinct research question and will therefore be presented as an independent study. Finally, this study focused on Korean older adults (≥65 years) within the context of a unique national health insurance system and demographic structure. Therefore, caution is warranted when generalizing these findings to younger age groups or to populations in different sociocultural and healthcare settings.”